# Direct Cardiac Epigenetic Reprogramming through Codelivery of 5′Azacytidine and miR-133a Nanoformulation

**DOI:** 10.3390/ijms232315179

**Published:** 2022-12-02

**Authors:** Priyadharshni Muniyandi, Vivekanandan Palaninathan, Tatsuro Hanajiri, Toru Maekawa

**Affiliations:** 1Graduate School of Interdisciplinary New Science, Toyo University, 2100 Kujirai, Kawagoe 3508585, Saitama, Japan; 2Bio-Nano Electronics Research Centre, Toyo University, 2100 Kujirai, Kawagoe 3508585, Saitama, Japan

**Keywords:** nanovectors, non-viral gene therapy, epigenetic reprogramming, direct reprogramming, 5′Azacytidine, MicroRNAs

## Abstract

Direct reprogramming of cardiac fibroblasts to induced cardiomyocytes (iCMs) is a promising approach to cardiac regeneration. However, the low yield of reprogrammed cells and the underlying epigenetic barriers limit its potential. Epigenetic control of gene regulation is a primary factor in maintaining cellular identities. For instance, DNA methylation controls cell differentiation in adults, establishing that epigenetic factors are crucial for sustaining altered gene expression patterns with subsequent rounds of cell division. This study attempts to demonstrate that 5′AZA and miR-133a encapsulated in PLGA-PEI nanocarriers induce direct epigenetic reprogramming of cardiac fibroblasts to cardiomyocyte-like cells. The results present a cardiomyocyte-like phenotype following seven days of the co-delivery of 5′AZA and miR-133a nanoformulation into human cardiac fibroblasts. Further evaluation of the global DNA methylation showed a decreased global 5-methylcytosine (5-medCyd) levels in the 5′AZA and 5′AZA/miR-133a treatment group compared to the untreated group and cells with void nanocarriers. These results suggest that the co-delivery of 5′AZA and miR-133a nanoformulation can induce the direct reprogramming of cardiac fibroblasts to cardiomyocyte-like cells in-vitro, in addition to demonstrating the influence of miR-133a and 5′AZA as epigenetic regulators in dictating cell fate.

## 1. Introduction

Myocardial infarction (MI) is the most prevalent cause of cardiovascular disorder that prompts irreparable damage to the heart that leads to death [1]. MI progression includes depletion of oxygen to heart muscles, which induces the death of heart muscles, and the formation of stiff scar tissues to replace the lost tissue (cardiac remodeling) that is critical and challenging to repair [2]. Despite advances in cellular and non-cellular therapies to prevent cardiac remodeling due to a heart attack, the mortality in cardiovascular disease still remains overwhelming [3]. In order to overcome this challenge; direct reprogramming is a promising strategy that fosters the possibility of changing the somatic cells to direct patient-specific cell bypassing stemness. This approach can eliminate the limitations of stem cells and allow precision medicine in regenerative medicine. However, due to the constraints of current delivery strategies and epigenetic barriers, the yield of reprogrammed new cardiomyocytes from cardiac fibroblasts is low. Thus, to efficiently reprogram commercially available adult cardiac fibroblast, we anticipate the global changes in epigenetic marks such as DNA methylation is critical in cardiovascular diseases because often abnormal methylation of CpG islands leads to gene expression modification [4,5].

Epigenetic modifications lead to the alteration of gene expression where the underlying genetic sequence is not changed. For instance, DNA methylation is an important form of epigenetic regulation, where the addition of a methyl group from S-adenosyl-L-methionine (Sam) to 5′cytosine of CpG dinucleotide limits the entry of transcription machinery to the promoter region leading to gene silencing and chromosome inactivation [6]. Several reports have demonstrated the key epigenetic marks in cardiovascular diseases. For instance, Zhou et al. demonstrated that silencing of Bmi1 facilitates removing the histone H2A lysine at the cardiac gene, increasing reprogramming efficiency [7]. Similarly, Testa et al. showed that 24 h treatment of CFs or MEFs with Bmi1 inhibitor PTC-209 repressed inflammatory signaling such as STAT3, IL-6, and ERK1/2 and increased reprogramming efficiency by ~25% [8]. In another study, Singh et al. reported that HDAC inhibitor alone or combined with WNT inhibitor improved reprogramming efficiency [9]. While Hyun et al. showed that epigenetic writers, erasers, and readers play an important role in controlling the changes in epigenetic landscapes [10]. It was interesting to note that inhibiting EZH2 methyltransferase activity doubled the number of beating cardiomyocytes [11]. EZH2 expression could be downregulated by the overexpression of miR-1, miR-133, miR-208, and miR-499 in neonatal CFs; however, it upregulated Kdm6a [12]. On the contrary, double knockdown of Kdm6a and Kdm6b inhibited the upregulation of Gata4, Mef2c, and Tbx5 mRNA expression, leading to an impaired reprogramming process [13].

Several reports are available on histone modification; however, only limited reports on DNA methylation for direct cardiac reprogramming (DCR). Thus, how these epigenetic changes are regulated remains challenging to comprehend [14]. To overcome the current limitation, we aim to deliver the epigenetic modifier 5′AZA and miR-133a using different presentation strategies. 5′AZA is a DNA methylase inhibitor, and its analog 2′-deoxycytidine was first synthesized in 1964 [15]. Previous reports on 5′AZA-induced differentiation of mesenchymal stem cells to cardiomyocytes [16,17,18] have made efforts to understand how epigenetic modification affects cardiomyocyte differentiation [19]. However, the DCR of cardiac fibroblasts to cardiomyocytes using 5′AZA induction has never been explored.

The human heart expresses more than 800 miRNAs, and among these, miR-133a is the most abundant type in the human myocardium [20,21,22]. miR-133a deletion results in impaired cardiac development at embryonic and postnatal stages. Further, the inhibition of only miR-133a led to cardiac hypertrophy in adult mice [23]. The most common pathological remodeling in the human heart is fibrosis and hypertrophy. miR-133a is well known to reduce the severity of both fibrosis and hypertrophy [23]. Anyway, it is adequately described in several studies that miR-133a is cardioprotective [24,25,26]. It is noteworthy that miR-133a also regulates DNA methylation by inhibiting Dnmt-1, Dnmt-3a, and Dnmt-3b [24]. Moreover, In human mesenchymal stem cells, miR-133a promoted cardiogenic differentiation by targeting epidermal growth factor receptors [27]. Cardiac differentiation is crucial for repairing and regenerating the myocardium and restoring the damaged cardiomyocytes.

Due to lack of oxygen, stiff scar tissue is produced to replace the dead myocytes with the cardiac fibroblasts in a failing heart. This causes the accumulation of fibrosis and, in turn, causes fibrosis. Cardiac fibroblast, which replaces the dead cardiomyocyte with stiff scar tissue, can be reprogrammed into cardiomyocytes using a cocktail of miRs such as miR-1, 133a, 208, 499 in mice [28]. In a damaged heart, the transdifferentiation of fibroblasts to iCMs helps in the reversal of pathological remodeling.

In our previous study, we found that nanoparticle or scaffold-mediated delivery of reprogramming cocktail invitro could help in DCR of cardiac fibroblast to cardiomyocyte-like cells with a minimal dose of reprogramming miR cocktail, including miR-1 and miR-133a [29,30]. Therefore, considering the role of miR-133a in epigenetic modification, fibrosis, and the earlier reports on 5′AZA, we hypothesize that demethylation is an essential epigenetic factor that could enhance DCR. Thus, we aim to co-deliver a microRNA and a small molecule as a reprogramming cocktail using Poly (D, L-lactic-co-glycolic acid) (PLGA) NPs. PLGA is an FDA-approved polymer that has demonstrated potential in drug/gene delivery in nanomedicine [31]. Further, PLGA is a widely used biopolymer due to its excellent biocompatibility, biosafety, and biodegradability [32]. In this study, we encapsulated PEI-miR-133a and 5′AZA in a PLGA nanoparticle for epigenetic reprogramming. We anticipate that the muscle-specific miRNA 133a, which has a crucial role in cardiac development and differentiation, would enhance cardiac fibroblast reprogramming potential with a DNA methylation inhibitor 5′AZA that would control the signature of the cell. Thus, by combining an epigenetic regulator and a genetic regulator, we demonstrate that DNA targeting can selectively remove the epigenetic barrier and increase the reprogrammed cells, accentuating the role of the epigenetic barrier in reprogramming.

## 2. Results and Discussion

### 2.1. Synthesis and Characterization of PEI-miR-133a, PLGA, PLGA-AZA, PLGA-PEI-miR-133a, PEI-miR-133a-AZA PLGA Nanoparticles

The PEI-miR-133a complex was formed by a self-assembly mechanism. A cationic material branched polyethyleneimine PEI_25k_ is used to complex miR-133a before encapsulating it into PLGA nanoparticles. PEI_25k_ is used in this study as it confers superior gene complexing ability and high transfection efficiency [33]. The cationic PEI_25k_ and miR-133a complexes were formed at an N:P ratio of 1:10. The morphology of PEI-miR-133a polyplexes was evaluated using TEM (Figure 1a), which showed the spherical morphology of the polyplexes. Further, the polyplexes with aggregated individual spheres had an average zeta size of 124 nm (Figure 1b). The successful complexation was confirmed by the shift in the surface zeta potential to a positive surface charge (Figure 1c). Further, the gel retardation assay (Appendix A) showed that miR-133a was present at the native position, and the band did not migrate from the well, thus confirming the successful polyplexes formation. Further, The PLGA nanoparticles (void, AZA drug, and PEI-miR-133a-AZA) were synthesized using the double-solvent evaporation method. The matrix of PLGA NPs is formed by the interactions between the hydrophobic polyvinyl alcohol (PVA) groups with the PLGA chain and hydrophilic PVA groups with the water phase [34]. The scanning electron microscopy (SEM) images showed that the PLGA nanoparticles had a smooth morphology with an average mean size of 198 nm for void particles (Figure 2a) and 298 nm for loaded PLGA-PEI-miR-133a-AZA (Figure 2b), respectively. PLGA-PEI-miR-133a, PLGA-AZA, and PLGA-PEI-miR-133a-AZA particles showed a polydispersity distribution (PDI) between 0.063–0.032 (Table 1). The NPs possessed a ζ potential of −15.8 to −15.6 mV. The ζ potential of PLGA-void (Figure 2c) and PLGA-PEI-miR-133a-AZA NPs (Figure 2d) did not change significantly owing to the highly anionic nature of miR-133a. Hence, indicating a successful miR encapsulation within PLGA NPs. PLGA NPs had a negative surface charge imparted by PVA surfactant due to the physical entrapment within the surface layers of the polymer. We used dimethyl sulfoxide (DMSO) as a cosolvent to dissolve the 5′AZA drug to ensure high loading and encapsulation efficiency. The result shows high encapsulation of miRNA polyplexes, the drug, and the co-encapsulation of AZA-PEI-miRNA into PLGA, as shown in Table 1. Similarly, high encapsulation was observed with the codelivery of the drug and miRNA using PLGA as a vector [35]. The percentage yield of the nanoparticles was 37% on average. The encapsulation efficiency yield and loading efficiency were consistent during the synthesis of each batch of PLGA NPs.

FTIR is a powerful analytical tool available today that can be used to analyze any sample, from liquids and pastes to powders, thin films/fibers, gases, and surfaces [36]. In order to verify the successful encapsulation of the target small molecule and genetic component inside PLGA Ns, FTIR spectra were recorded. FTIR spectra confirmed characteristic PLGA polymer peaks with no additional new peaks related to the IR spectra of either the encapsulated drug or miRNA (Appendix A), which leads us to believe that the target cargo is successfully encapsulated inside the NPs. Refer to Appendix A for the characteristic peak positions and assignments for PLGA NPs that is comparable to a previous study [37].

### 2.2. Encapsulation and Stability of PEI-miR-133a Polyplexes

In order to evaluate the polyplexes encapsulation efficiency and stability, the supernatant obtained during PLGA NPs synthesis was analyzed by gel electrophoresis. The results showed that no visible miRNA bands were seen, thus confirming a high encapsulation efficiency of polyplexes within the PLGA NPs (Figure 3a). To evaluate the stability of PLGA NPs, the PLGA NPs were digested with SDS and loaded on a gel. The results confirmed the presence of polyplexes in the lane that possessed the digested PLGA NPs. Further, as seen in Figure 3b, miR-133a encapsulated inside the digested PLGA NPs (Lane 2) migrated to a band position similar to that of the naked miR-133a (Lane 4). This characteristic band confirms that the structure of nucleic acid is not damaged during synthesis and is successfully encapsulated inside the NPs.

### 2.3. Invitro Release Studies

The release of the therapeutic factors depends on several factors, such as temperature, type of NPs formulation, release medium, and pH [34]. Also, different events lead to the drug release kinetics in PLGA microspheres, including diffusion of the drug through the polymer matrix, polymer degradation, and erosion [38]. PLGA NPs displayed biphasic release profiles (burst release and sustained release) when tested for both pH 7.4 and pH 5.0, which mimic the pH of the extracellular and intracellular microenvironment, respectively. The release profiles were observed for 72 h under ambient conditions at 37 °C using ribogreen assay for miR-133a and UV absorbance at 234 nm for the 5′AZA. The initial burst release may be due to the leakage of any 5′AZA and miR-133a present near the surface of the PLGA NPs. The subsequent release from the particles depends on diffusion via the polymeric matrix and the water-filled pores. Initial burst release after 1 h was 9.4% and 9.7% for miRNA and 10.5% in pH 5.0 and 12.9% for the drug in pH 7.4 and pH 5, respectively. After the initial burst release, the sustained, steady slow release was observed. Most of the miR-133a and 5′AZA release was observed between 24 to 48 h. The release profile was slow after 24 h because of the polymer content and its impact on decelerating drug/miR release as a result of the increase in the particle size and reduced surface area available for 5′AZA and miR-133a release. In the case of pH 5.0, the release percentage of miRNA has increased from 45–70%, whereas 40–78% was observed in the case of pH 7.4. However, drug release was rapid, with a release percentage of 60–70% after 48 to 72 h, followed by a steady, stationary phase, Figure 4. The accelerated drug release in the final stages of incubation could be associated with several factors, including swelling of NPs, the introduction of large pores or cracks on NPs surfaces due to polymer degradation, or particle disintegration [38]. The release profiles of PLGA NPs in this study are coherent with the previous reports on PLGA nanocarriers for biomedical applications [29,39].

### 2.4. Cellular Uptake

FITC-labeled fluorescent PLGA NPs were added to HCF and HCM cells to determine the intracellular internalization of nanoparticles inside the cells [40]. Similarly, the intracellular internalization of cy5 labeled miR-133a was also observed after addition to HCF and HCM cells and incubating for 4 h. As seen in Figure 5, irrespective of the cell line, an efficient particle initialization was observed in both HCF and HCM. Similar results were observed with Cy5 labeled miR-133a uptake (Appendix A).

The green-stained cells represent PLGA-FITC NPs, the red-stained cells represent Cy5-miR-133a uptake, and the blue represents Nucblue. The confocal images confirmed that NPs were seen inside the cells and not simply absorbed to the outer surface. The efficient uptake is due to the average size of the PLGA NPs. Similar uptake was reported previously in different cell lines using PLGA NPs [29,41]. As previous reports suggest that the downsizing of nanoparticles results in the increment of the surface area that, in turn, enhances the contact with the cell membrane. Thereof, the particles with smaller size have superior interaction with the cell surface [42]. These results are suggestive that these particles are compatible with in vivo applications by increasing blood circulation interaction. The results clearly show that the particles were quickly endocytosed with high particle uptake, and a high number of positive cells can be achieved upon codelivery of miRNA and AZA. In order to further quantitatively confirm the uptake, flow cytometry analysis was used. In coherence with our CSLM results, we observed a high particle uptake (Appendix A). The fluorescent signal of FITC was clearly shifted after incubation of 4h. These results further confirm the application of PLGA NPS as a promising drug delivery system in regenerative medicine due to its ideal size [43].

### 2.5. Cytocompatibility and Cell Viability of PLGA NPs

The biocompatibility assay included analyzing all the synthesized PLGA NPs, in addition to free 5′AZA and PEI-miR-133a polyplexes. Cells treated with varying void concentrations of nanoparticles were incubated for 3 days, where the cell viability was observed to be >80% for all concentrations used (Figure 6). This could be attributed to the fact that PLGA is an FDA-approved biodegradable and biocompatible polymer. These results are consistent with the reports on the biocompatibility of PLGA polymer [44].

Next, the cytotoxicity effects of the PEI-miR-133a, AZA, PEI-miR-133a-AZA loaded PLGA, polyplexes, and free 5′AZA were tested using presto blue assay. The HCF and HCM cells were incubated for 24, 48, and 72 h with PEI-miR-133a, AZA, PEI-miR-133a-AZA loaded PLGA at a concentration of 100 μg/mL in comparison with miR-133a polyplexes (50nM was used to for complex with PEI at an N:P ratio 10) and free drug (1 mg/mL) for evaluating the percentage cell viability. As shown in Figure 7 and Appendix A, an average cell viability percentage of 80% was observed in all the cells treated with the PEI-miR-133a, AZA, PEI-miR-133a-AZA loaded PLGA NPs, which indicates that the designed nanoparticles physicochemical properties have not triggered any biological response.

Although the addition of polyplexes has no significant change in cell viability in comparison with different PLGA NPs, however, it is evident from Figure 7 and Appendix A that the addition of free 5′AZA to the cells has exhibited reduced cell viability due to the high concentration of the free drug [45]. The cell viability of the free AZA was an average of 72% at all time points. Collectively, the cytotoxicity results confirm that PLGA NPs exhibited minimal cytotoxicity in comparison to the control. The results demonstrate that PLGA-mediated drug delivery system (DDS) possesses an excellent safety profile which may gain promising grounds as an alternative for immunogenic and toxic viral vectors in regenerative medicine.

### 2.6. Live/Dead Cytotoxicity Analysis

A live-dead assay was performed to affirm the quantitative results of cytotoxicity. The HCF cells were added with 100 mg/mL of target cargo-loaded PLGA NPs (AZA, PEI-miR-133a, PEI-miR-133a-AZA) for 72 h. Following the treatment, cells were stained with live/dead reagents as per the manufacturer’s instructions. Figure 8 gives us visual evidence of live/dead assay with live cells stained in green. The live-dead analysis further manifests that PLGA formulation is less toxic to cells, in agreement with the cytotoxicity results. In the images, Figure 8(b1–d1), most cells remain alive and adhere to the culture plate with distinct morphology resembling the control group (Figure 8(a1)). No dead cells were observed in all treatment groups of PLGA NPs, as evident from the respective images for AZA, PEI-miR-133a, and PEI-miR-133a-AZA. The data of cytotoxicity and live dead analysis supports the claim that PLGA is biocompatible, and the metabolic activity of the cells is not affected even after 72 h treatment.

### 2.7. cTnT Expression and Global Methylation Analysis

To demonstrate the DCR of cardiac fibroblasts, we co-delivered miR-133a, a muscle-specific synthetic miRNA mimic, and 5′Azacytidine, a DNA methylation inhibitor using PLGA NPs.

The cells were analyzed using late cardiac marker cTnT on day 7. Parallelly, 5′-methyl-2′-deoxycytidine levels were quantified using global methylation ELISA on day 3 and day 7 to prove that miR-133a and 5′AZA act as epigenetic modifiers to support the epigenetic DCR of HCF cells to adult iCMs such as cells. Previously the role of miR-133a and 5′AZA in epigenetic modification is adequately explored [24,46]. More specifically, miR-133a has been recognized to promote DCR in past investigations, including us [28,29,47].

In the present study, our results confirmed that upon treatment of encapsulated PLGA NPs (AZA, PEI-miR-133a, and PEI-miR-133a-AZA) with HCFs, the cells were reprogrammed to cardiomyocyte-like cells in vitro as opposed to the untreated cells. The effect of PLGA-mediated miR-133a, Aza, and miR-133a transfection on AHCFs trans-differentiation into iCMs was evaluated at different time points by different analyses depicted in Figure 9A. As a negative control, AHCFs were also transfected with PLGA-Neg-mimic. The morphological analysis confirmed aggregation, orientation, and fusion of cells as early as day 3 in PLGA-AZA-treated and PLGA-PEI-miR-133a-AZA-treated cells in comparison with PLGA-PEI-miRNA only treated cells. Therefore, we believe that 5′AZA as a demethylating agent reduces the length of the transdifferentiation process. In addition to the morphological change, we also observed cTnT, a late cardiac marker expression following 7 days of induction with PLGA-AZA, PLGA-PEI-miR-133a, and PLGA-PEI-miR-133a-AZA. It was noteworthy that PEI-miR-133a alone (Figure 9B(b1–b4)) was efficient in reprogramming the HCF cells. The results were similar to our previous report on dual miR for reprogramming HCF cells in vitro [29]. Further, it is demonstrated in earlier reports that miR in combo or miR alone could induce DCR [28]. Similarly, Li et al. showed that miR combo reprogrammed neonatal murine cardiac fibroblast to iCMs [47]. Similarly, 5′AZA alone had cTnT positive cells (Figure 9B(c1–c4)) similar to the miR-133a treatment group. These results were comparable to the findings of sun et al., where they reprogrammed rat bone marrow (BM) derived very small embryonic-like stem cells (VSELs) in differentiating to cardiomyocyte-like cells invitro using 5′AZA treatment alone [48]. In another study, mouse P19 embryonic carcinoma (EC) cells were differentiated into cardiomyocytes with 5′AZA treatment by epigenetic coregulation and FRK signaling [49]. However, a significant number of cTnT-positive cells (29%) were found in the experimental group of cells treated with a cocktail of miR-133a and 5′AZA (Figure 9C. qPCR analysis showed that miR-133a, AZA, and miR-133a-AZA transfected cells upregulated the expression of early cardiac transcription factors (TFs), with a significantly increased expression of GATA4 (*p* = 0.007), MEF2C (*p* = 0.0004), TBX5 (*p* < 0.0001) and HAND2 (*p* = 0.0001) cardiac TFs, compared to negmiR transfected cells, 3 days after transfection Figure 9D–F. Additionally, the expression of NKX2.5 cardiac TF was increased in miRcombo transfected cells compared to controls, although not significantly Figure 9F. These results were further supported by global methylation ELISA quantification of 5′-methyl-2′-deoxycytidine. The results showed a reduced 5′-methyl-2′-deoxycytidine value compared with the control, which suggests a possible correlation between epigenetic modification and DCR. As shown earlier, miR-133a has regulated DNA methylation in the diabetic heart [24]. Similarly, our data demonstrated that miR-133a and 5′AZA could enhance the reprogramming of cardiac fibroblasts by DNA hypomethylation. The present in vitro results confirm the idea that the miR-133a and 5′AZA are key epigenetic erasers that could regulate DNA methylation to dictate cell fate.

DNA methylation occurs when a methyl group is covalently added to the 5th carbon of the cytosine ring by DNA methyltransferases (DNMT), which results in 5-MedCyd. DNA methylation is the major epigenetic modification. In somatic cells, 5-MedCyd is found in the context of paired symmetrical methylation of dinucleotide CpG, while in embryonic stem cells, the majority of 5-MedCyd is observed in the non-CpG context. Therefore, to evaluate the reprogramming efficiency of cardiac fibroblasts using epigenetic modifiers, ELISA-based global methylation analysis of cells treated with PLGA NPs (PEI-miR-133a, AZA, PEI-miR-133a-AZA) was carried out on day 3 and day 7. The results showed a reduction in DNA methylation on both selected time points Figure 10a,b. Nevertheless, compared to day 3 (Figure 10b), a significant reduction in 5-MedCyd was observed on day 7 (Figure 10b). Further, on day 7, PLGA-5′AZA (23%) treated cells show a slightly reduced 5-MedCyd value compared to the PLGA-PEI-miR-133a (26%). While the 5-MedCyd value of PLGA-PEI-miRNA and PLGA-5′AZA were significantly reduced, the co-delivery of miR-133a and 5′AZA (20%) treated cells showed further hypomethylation, which could be due to the inhibition of DNA methylation activity by both miR-133a and 5′AZA.

## 3. Materials and Methods

### 3.1. Materials

Branched PEI 25kDa, Acid terminated poly (D, L-lactic-co-glycolic acid) (PLGA) (MW: 7000–17,000, 50:50), 5′Azacytidine (AZA), polyvinyl alcohol (PVA), MW 31–50 kDa, 87–89% hydrolyzed, fluorescein isothiocyanate (FITC) (Sigma Aldrich, Japan), Fibroblast medium 3, detach kit, Adult Human Cardiomyocyte (HCM) (PromoCell), Myocyte growth medium (Promocell), Adult Human Cardiac Fibroblasts (HCF) (PromoCell), PrestoBlue (ThermoFisher Scientific), pre-miR^TM^ miRNA precursors for miR-miR-133a (Ambion), Quant-iT Ribogreen RNA Assay Kit (Thermo Scientific), 2′,7′-Dichlorofluorescin diacetate (Sigma Aldrich, Japan), Invitrogen^TM^ Purelink^®^ Genomic DNA kit (K182001), Invitrogen^TM^ Live/Dead cytotoxicity kit, Lipofectamine 2000, mouse monoclonal anti-cardiac Troponin T (cTnT) antibody, Donkey anti-mouse Alexa fluor 488 (ThermoFisher Scientific) and Fixation buffer and Perm/Wash buffer (BD Bioscience). All other chemicals or reagents were of an analytical grade acquired either from Sigma (Merck) or Wako Chemicals, Japan. NucBlue Live Ready Probes Reagent was procured from Thermo Fisher Scientific. Applied Biosystems TaqMan Gene Expression assays (FAM-labeled) against the genes GATA4, TBX5, HAND2, MEF2c, NKx2.5, and GAPDH were purchased from Thermo Fisher Scientific.

### 3.2. Methods

#### 3.2.1. miRNA Polyplexes Preparation and Gel Retardation Assay

The PEI miRNA complexes were synthesized by a previously established method [29]. Briefly, PEI and miRNA were complexed by the self-assembly method. The PEI polymer solution of concentration 1 mg/mL was added to miRNA at a concentration of 60 pmol at nitrogen to phosphate ratio of 1:10. The mixture of polymer: miRNA was incubated at room temperature for 30 min for the polyplexes formation. The formation of polyplexes was confirmed by gel retardation assay. The polyplex solution of 10 μL was mixed with 4 μL of loading buffer loaded onto 1% agarose gel with gel red—Tris-acetate (TAE) running buffer of pH 8.3 and electrophoresed at 100 V for 30 min. The presence of miRNA bands was visualized with an ultraviolet illuminator and photographed using gel doc (Image quant LAS 4000).

#### 3.2.2. Synthesis and Characterization of PEI-miR-133a, PLGA AZA, and PLGA-PEI-miRNA-AZA

The AZA drug, PEI-miR-133a, and miR-133-AZA were encapsulated in PLGA by a double emulsion solvent evaporation method. In this method, 30 mg of PLGA was dissolved in 1 mL of DCM. The dissolved polymer solution was kept under magnetic stirring for complete dissolution. Briefly, 1 mg of the drug was added to 1 mL of DMSO. The mixture was added to an aqueous solution of PVA with 5% (*w*/*v*) and emulsified using a probe sonicator for 3 min on the ice. In order to evaporate the dichloromethane; the emulsion was agitated using a magnetic stirrer for 3 h. Then it was centrifuged for 30 min at 6000 rpm with subsequent washing with RNAase-free water 3 times to remove the traces of PVA. The wash solution was collected to calculate free NPs in supernatants, which are not encapsulated. The collected nanoparticles were lyophilized for 24 h and stored at −20 °C until further use. Simultaneously, the same method was used to synthesize miRNA-loaded particles and AZA-miRNA-loaded particles. A similar protocol was used with 50 nM of miRNA complexed with polyethyleneimine with an N:P ratio of 1:10 and added to PLGA polymer solution dissolved in DCM. In this case, AZA-miRNA, 1 mg of the drug, and 50 nM of miRNA complexed with polyethyleneimine with an N:P ratio of 1:10 was added to 1 mL polymer solution. The void nanoparticles are prepared similarly without polyplexes and FITC.

#### 3.2.3. Physico-Chemical Characterization

The size and morphology were analyzed using transmission electron microscopy (TEM) (JEOL JEM-2100 TEM, Tokyo, Japan). To visualize the miRNA formation previously established negative staining method was used [29]. Polyplexes at an N/P ratio of 10 was prepared by adding the desired volume of PEI polymer solution of 1mg/mL and incubated for 30 min. A 10 μL of the sample was dropped on the TEM carbon-coated copper. Following the treatment, Ti blue dye was added to the polyplexes solution on the grid, and the grid was allowed to dry at room temperature. TEM was used to characterize the morphology of PEI-miRNA polyplexes at an accelerating voltage of 100 kV. Scanning electron microscopy (SEM) (JEOL, JSM-7400, Tokyo, Japan). and TEM (JEOL JEM-2100 TEM, Tokyo, Japan) were used to evaluate the size and the morphology of synthesized NPs. Briefly, the nanoparticles were suspended in RNA-free water and dried for 24 h. The air-dried samples were platinum-coated, and the SEM images were acquired with an accelerating voltage of 5 kV and a beam current of 20 µA. Similarly, 10 μL of the sample was dropped on the hydrophilized Cu microgrid and air-dried at room temperature, and TEM images were acquired with an accelerating voltage of 100 kV.

The hydrodynamic diameter and ζ-potential measurements were carried out using Zetasizer (Malvern, Nano-ZS) in triplicates. The average size and polydispersity index (PDI) of the nanoparticles and polyplexes were measured using disposable polystyrene cuvettes. Zeta potential measurements were carried out using dip cells. In the case of polyplexes, the particles were diluted with RNase-free water to 1 mL volume before measurement. Fourier Transform Infrared (FTIR) analysis was performed using Nicolet iS50/Raman, Thermo Scientific Spectrometer, where the IR spectra were recorded over a region of 4000–400 cm^−1^ with 4 cm^−1^ resolution.

#### 3.2.4. Percentage Yield and Encapsulation Efficiency

The amount of miRNA loaded into the nanoparticles (NPs) was determined by a previously reported method [29]. The encapsulation efficiency was determined by the number of untrapped miRNA mimics in the wash solution quantified by Quanti-iT Ribogreen RNA Assay. The fluorescence intensity resulting from the miRNA binding to ribogreen reagent was determined using a microplate reader. Similarly, In the case of the drug, the absorbance was measured using UV spectroscopy. miRNA/drug loading in nanoparticles was determined by subtracting the total amount of miRNA/drug recovered in the wash solution miRNAw/drugw from the initial amount of miRNAi/drugi added. The encapsulation efficiency was calculated using the following equation [1], the loading efficiency was calculated using the following formula [2], and the yield of nanoparticles was calculated using the following equation [3],
(1)$$\mathrm{Encapsulation}\;\mathrm{Efficiency}=\frac{\mathrm{miRNA}/\mathrm{drug}\;i-\mathrm{miRNA}/\mathrm{drug}\;w\;}{\mathrm{miRNA}/\mathrm{drug}\;i}100$$
where miRNA/drug i is the initial amount of miRNA/drug, and miRNA/drugw is the amount of miRNA or drug in the wash solution.
(2)$$\%\;\mathrm{loading}=\frac{\mathrm{amount}\;\mathrm{of}\;\mathrm{miRNA}/\mathrm{Drug}\;\mathrm{in}\;\mathrm{the}\;\mathrm{particle}\;}{\mathrm{gross}\;\mathrm{weight}\;\mathrm{of}\;\mathrm{nanoparticles}}100$$
(3)$$\%\;\mathrm{yield}=\frac{\mathrm{dry}\;\mathrm{weight}\;\mathrm{of}\;\mathrm{the}\;\mathrm{Nanoparticle}\;\mathrm{obtained}\;}{\mathrm{WmiRNA}/\mathrm{drug}\;+\;\mathrm{WPLGA}}100$$
where Wm is the amount of miRNA/drug, and WPLGA is the amount of PLGA used for nanoparticle synthesis.

#### 3.2.5. Gel Electrophoresis Assay to Confirm Encapsulation and Stability of miRNA Polyplexes to PLGA Nanoparticles

The confirmation of encapsulation and stability of miRNA polyplexes to PLGA NPs was confirmed by agarose gel electrophoresis. In order to confirm the encapsulation of PEI-miRNA, supernatant obtained during the synthesis of PLGA NPs were gel electrophoresed to confirm the high encapsulation. Further, to quantitatively analyze the stability, briefly, the nanoparticles were centrifuged, and the pellet was ruptured with an aqueous solution of 1% (*w*/*v*) sodium dodecyl sulfate (SDS) to determine the presence of miR-133a qualitatively. To each 10 μL sample, 2 μL loading buffer was added, and the complexes were then loaded on 1 % agarose gel containing Gel Red immersed in Tris-acetate (TAE) running buffer (pH 8.3) and electrophoresed at 100 V for 30 min. The presence of miRNA bands was visualized with a UV illuminator and photographed using a gel doc imaging system (Image quant LAS 4000). Naked miRNA was used as a control.

#### 3.2.6. Invitro miRNA/Drug Release

Invitro miRNA and drug release profiles from PLGA NPs were assessed at pH 7.4 and pH 5.0 to simulate extra-cellular (pH 7.4) and intra-cellular (pH 5.0) microenvironments. Briefly, 15 mg of NPs were dispersed in 15 mL of PBS buffer, and 1 mL was aliquoted in 2 mL Eppendorf tubes with a final concentration of 1 mg/mL. Tubes were incubated with the agitation of 120 rpm for 3 days at 37 °C. At predetermined intervals, the tubes were centrifuged at 15,000 rpm for 30 min. For the drug, the absorbance was measured at 234 nm, whereas miRNA release was estimated by Quanti-iT Ribogreen RNA, where a working solution of 100 μL was added to each well in 96 well plates containing 100 μL of standard, blank and unknown samples that were briefly mixed and incubated for 5 min at room temperature (protected from light). miRNA concentration of each sample was measured by a microplate reader (Power scan HT microplate reader, Dainippon Sumitomo Pharma, Japan). The release percentage was calculated using the following equation [5],
(4)$$\%\;\mathrm{release}=\frac{\;\mathrm{Released}\;\mathrm{miRNA}/\mathrm{drug}\;}{\mathrm{Total}\;\mathrm{miRNA}/\mathrm{drug}}100$$
where ‘Released miRNA/drug’ is the concentration of the miRNA/drug released from the NPs and ‘Total miRNA/drug’ is the amount of miRNA/drug encapsulated in the NPs.

#### 3.2.7. Cell Culture

Human Cardiac Fibroblasts (HCF) or Human Cardiomyocytes (HCM) were cultured at a cell density of 5 × 10^6^ cells in a specialized Fibroblast medium 3 (Promocell) and Myocyte growth media (Promocell) respectively at 37 °C in a 5% CO_2_ incubator. The cells were subcultured every 6–7 days or until confluent.

#### 3.2.8. PLGA-PEI-miR133a Transfection

HCF was seeded at a cell density of 5 × 10^4^ cells in 6 well plates. After 24 h, cells were transfected with pre-miR^TM^ miRNA precursors for miR-133a using PLGA NPs encapsulated with polyplexes of miR-133a. Following the transfection with miR-133a, fresh media was added after 24 h incubation.

#### 3.2.9. PrestoBlue Cell Viability Assay

In order to evaluate the cytocompatibility of synthesized PLGA NPs, the adult human cardiac fibroblast (HCF) and human cardiomyocyte (HCM) cells were seeded in 96 well plates at a density of 5 × 10^4^ and maintained at 37 °C with 5% CO_2_. After 24 h of incubation, fresh media was replenished, and the NPs were added to cells in different concentrations (1 mg/mL, 750 mg/mL, 500 mg/mL, 250 mg/mL, 100 mg/mL). The cells were then incubated for 24, 48, and 72 h at 37 °C with 5% CO_2_. Similarly, miR-133a polyplexes (50nM was used for to complex with PEI at an N:P ratio 10), free drug (1 mg/mL) and PLGA void, PLGA NPs (PLGA-AZA, PLGA-PEI-miR-133a, PLGA-PEI-miR133a-AZA) were treated at a concentration of 100 mg/mL for 24, 48 and 72 h. The percentage of cell viability is estimated by the conversion of resazurin in the dye to fluorescent resorufin by the metabolically active cells. After the incubation, 10% of PrestoBlue reagent was added to each sample and incubated for 2 h. At the end of incubation, fluorescence intensity was measured at 530/590 nm using a microplate reader. The cell viability was calculated using equation [4],
(5)$$\%\;\mathrm{cell}\;\mathrm{viability}=\frac{\mathrm{Sample}\;}{\mathrm{Control}}100$$

#### 3.2.10. Live/Dead Cell Cytotoxicity Assay

The live/dead cytotoxicity of NPs was determined after treating the HCF cells with PLGA NPs. The cells were seeded on 35 mm confocal dishes at a cell density of 5 × 10^4^ cells for 72 h, and on the following day, PLGA void and PLGA NPs (AZA, PEI-miR133a, PEI-miR-133a-AZA) were added to the cells at a concentration of 100 μg/mL. Post-treatment, cells were washed with 1X PBS. The cells were then treated with dyes per the manufacturer’s instructions on the Invitrogen^TM^ Live/Dead cytotoxicity kit (L3224). The cells were imaged using the Confocal laser scanning microscopy (CLSM, Nikon A1+ Tokyo, Japan) at 495/515 nm for calcein and 528/617 for EthD-1.

#### 3.2.11. Cellular Uptake

CLSM analysis (CLSM, Nikon A1+ Tokyo, Japan) was carried out to analyze the NPs and miR-133a uptake by HCF and HCM. To assess the NPs and miR-133a uptake, FITC-PLGA NPs and Cy5 labeled miR-133a was used. The cells were cultured on a 35 mm glass-based culture dish at a seeding density 5 × 10^4^ for 24 h. Subsequently, 100 µg/mL concentration of FITC PLGA and PEI-Cy5miR polyplexes (50nM miR-133a complexed at an N:P ratio 10) were added to the cells and incubated for 4 h. The void PLGA was taken as control. After incubation of 4 h, the cells were rinsed with PBS pH 7.4 to remove the unbound nanoparticles. Then, the cells were analyzed at an excitation wavelength of 488 nm for FITC and 670 for Cy5. To further confirm the uptake of FITC update quantitively, after the incubation, HCF cells were trypsinized and resuspended in PBS (pH 7.4). Then the suspended cells were analyzed using a flow cytometer (Bay bioscience JSAN, Japan) and appsan software. Ten thousand events were read at an excitation wavelength of 488 nm (FITC) for each sample, and untreated cells were used as control.

#### 3.2.12. cTnT Cardiac Late Marker Expression

Adult human cardiac fibroblasts were transfected using scaffold-mediated in-situ delivery of miRNA. The transfected cells were maintained for 28 days. After day 7, the cells were trypsinized and fixed with a wash buffer. The cells were incubated with mouse monoclonal anti-cardiac Troponin T (cTnT) antibody at 1:200 dilution for 1 h at room temperature. Following incubation, the secondary antibody donkey anti-mouse Alexa fluor 488 was used at 1:200 respectively. After that, cells were washed with wash buffer and analyzed for cTnT expression using a confocal microscope.

#### 3.2.13. RNA Isolation Quantitative Real-Time PCR (qRT-PCR)

The cells were harvested after day 7 post-transfection to isolate the total RNA by RNeasy Micro kit, and cDNA was synthesized using Superscript-III reverse transcriptase. The cDNA template was used for qPCR analysis. The qRT-PCR was performed using TaqMan universal master mix, pre-designed TaqMan gene expression primer/probes assay against the genes GATA4, MEF2C, TBX5, HAND2, NKX2.5, and GAPDH. Real-time quantification system Applied Biosystems 7900 Fast Real-Time PCR system was used for the experiment. The TaqMan gene expression assay IDs of the respective target genes are GATA 4: Hs00171403_m1, MEF2C: Hs0398823_m1, TBX5: Hs00361155_m1, HAND2: Hs00232769_m1, NKX2.5:Hs03988823_m1, GAPDH: Hs02786624_g1. The relative mRNA expression levels were calculated and normalized relative to GAPDH mRNA as the internal control using the ∆∆Ct method.

#### 3.2.14. Global Methylation Analysis

Genomic DNA was extracted with Invitrogen^TM^ Purelink^®^ Genomic DNA kit (K182001) according to the manufacturer’s instructions. Briefly, the DNA was converted to single-stranded DNA by incubation at 95 °C for 5 min, followed by rapid chilling on ice. Samples were then digested to nucleoside by incubating the denatured DNA with nuclease P1 for 2 h at 37 °C in 20 mM sodium acetate (pH 5.2). Further, Alkaline phosphate was added and incubated for 1 h at 37 °C in 100 mM Tris (pH 7.5). After centrifugation, the supernatant was used for further ELISA assay using Global DNA Methylation ELISA Kit (5′-methyl-2′-deoxycytidine quantitation; CELL BIOLABS) according to the manufacturer’s protocol.

#### 3.2.15. Statistical Analysis

One-way ANOVA followed by Dunnett’s multiple comparisons test was performed using GraphPad Prism version 9.1.2.

## 4. Conclusions

This study shows the first evidence that co-delivery of miR-133a and DNA-methylation inhibitor 5′AZA enhance epigenetic mediated reprogramming. The role of DNA methylation in reprogramming is still poorly understood; however, it is well established that miR, in a combo or alone, can efficiently reprogram cells both in vitro [28,29,46] and in vivo [23,28,50]. Thus, we hypothesized that the use of non-viral DDS for the co-delivery of the drug and miR could improve biocompatibility while providing more insight into the regulation of epigenetic reprogramming. This study focuses on demonstrating the co-delivery of miR-133a and 5′AZA using PLGA NPs to promote epigenetic reprogramming by DNA hypomethylation. However, we have not explored the underlying molecular mechanism of direct epigenetic reprogramming using miR-133a and 5′AZA. Here, the synthesized PLGA NPs had an average size of 298 nm and exhibited a high encapsulation efficiency of 96% while showing a pH-dependent bi-phasic release of reprogramming factors. Further, the toxicity profile of the NPs demonstrated that this nanoformulation is highly cytocompatible on both HCF and HCM. Further, confocal microscopy results demonstrate a high cellular uptake of PLGA NPs.

In summary, the results of this study suggest that 5′AZA can enhance the reprogramming efficiency of HCF to iCMs by inhibiting DNA methylation. At the cellular level, 5′AZA inhibits the global methylation levels significantly when treated with PLGA-5′AZA; however, when co-delivered with PEI-miR-133a and 5′AZA loaded PLGA NPs, the inhibition of DNA methylation activity is further enhanced. Although the detailed mechanisms underlying these effects remain unclear. With the results of the proof of principle report on induction of HCF with miR-133a and 5′AZA, we propose that miR-133a and 5′AZA will be promising candidates for the epigenetic DCR of cardiac fibroblast to iCMs.

## Figures and Tables

**Figure 1 ijms-23-15179-f001:**
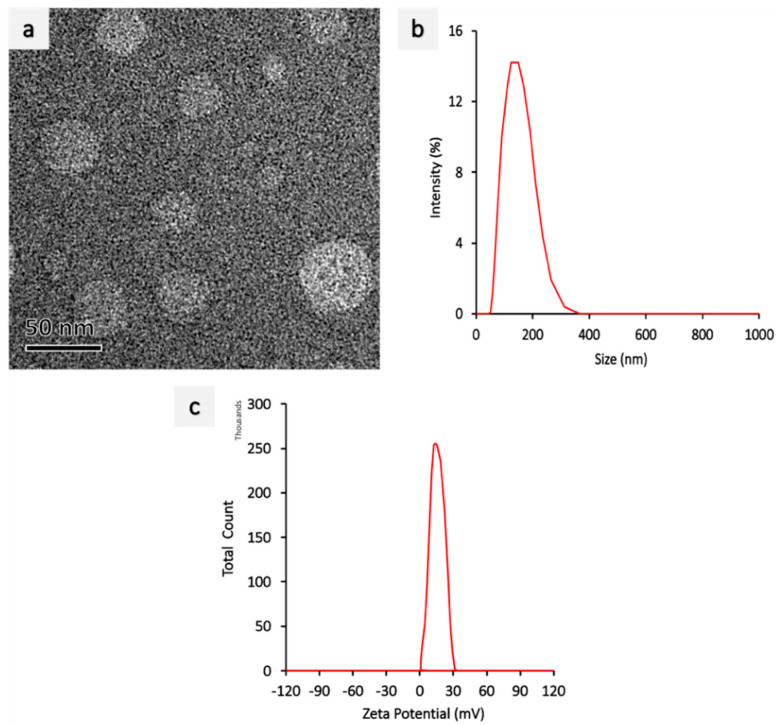
Polyplexes characterization (**a**) TEM micrograph showing spherical morphology of polyplexes (Scale bar = 50 nm). (**b**) polyplexes (1:10) size measurement by DLS, showing an average size of 124 nm. (**c**) zeta potential measure of polyplexes (1:10), showing positive zeta potential after forming polyplexes.

**Figure 2 ijms-23-15179-f002:**
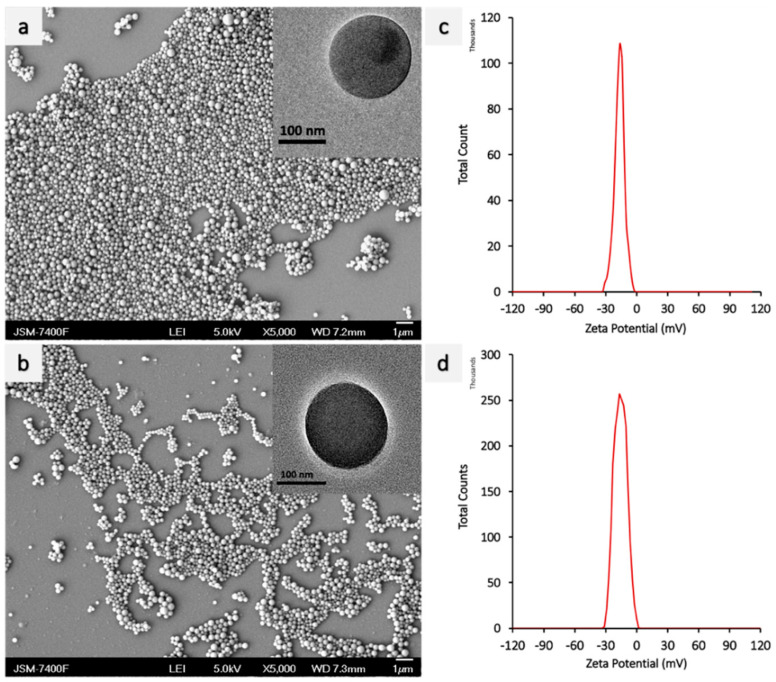
SEM image and Zeta potential of PLGA NPs (**a**) PLGA void (**b**) PLGA PEI-miR-133-AZA (**c**) zeta potential of PLGA void (**d**) zetapotential of PLGA PEI-miR-133-AZA.

**Figure 3 ijms-23-15179-f003:**
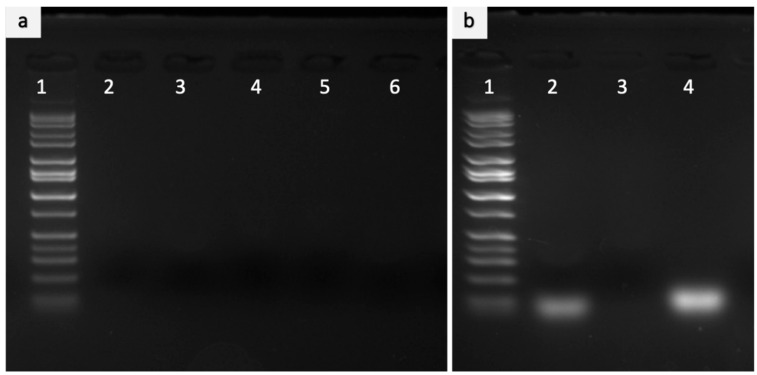
Evaluation of encapsulation and miRNA Stability inside the PLGA nanoparticles (**a**) Encapsulation of miRNA to PLGA NPs. Gel electrophoresis of the supernatant obtained during the synthesis of PLGA nanoparticles showed no miRNA, thus ensuring a high encapsulation of miRNA inside the PLGA. Lane 1: Marker, Lane 2–6: Supernatant. The gel image depicts that the structure of nucleic acids was not destabilized during the synthesis of nanoparticles. (**b**) Stability of miRNA in the PLGA nanoparticle. The pellet was disrupted with SDS to assess the encapsulation of miRNA. Lane 1: Marker, Lane 2: encapsulated miRNA, Lane 3: Blank Lane (Undigested PLGA NPs-Control) 4: Naked miRNA.

**Figure 4 ijms-23-15179-f004:**
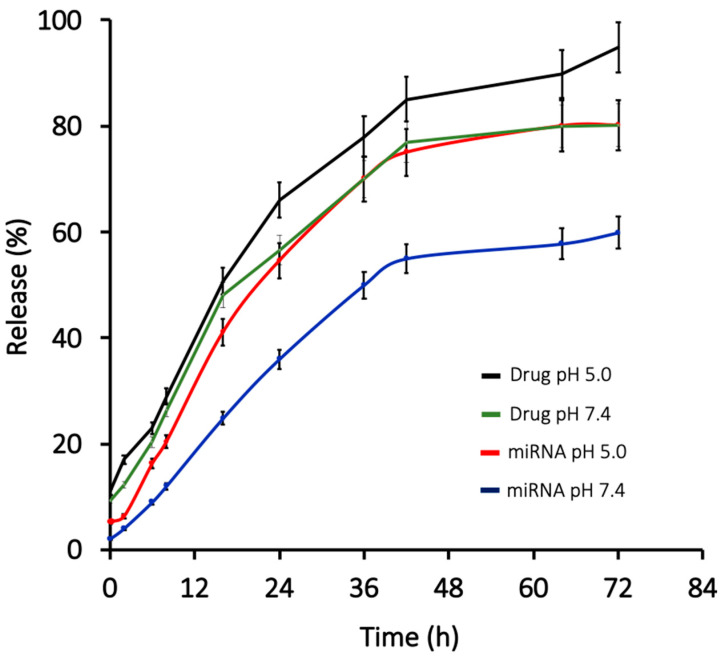
Invitro miRNA and drug release profiles from PLGA NPs were evaluated at pH 5.0 and pH 7.4 using ribogreen assay. The results suggest that both pH 5.0 and 7.0 showed biphasic release of miRNA and drug.

**Figure 5 ijms-23-15179-f005:**
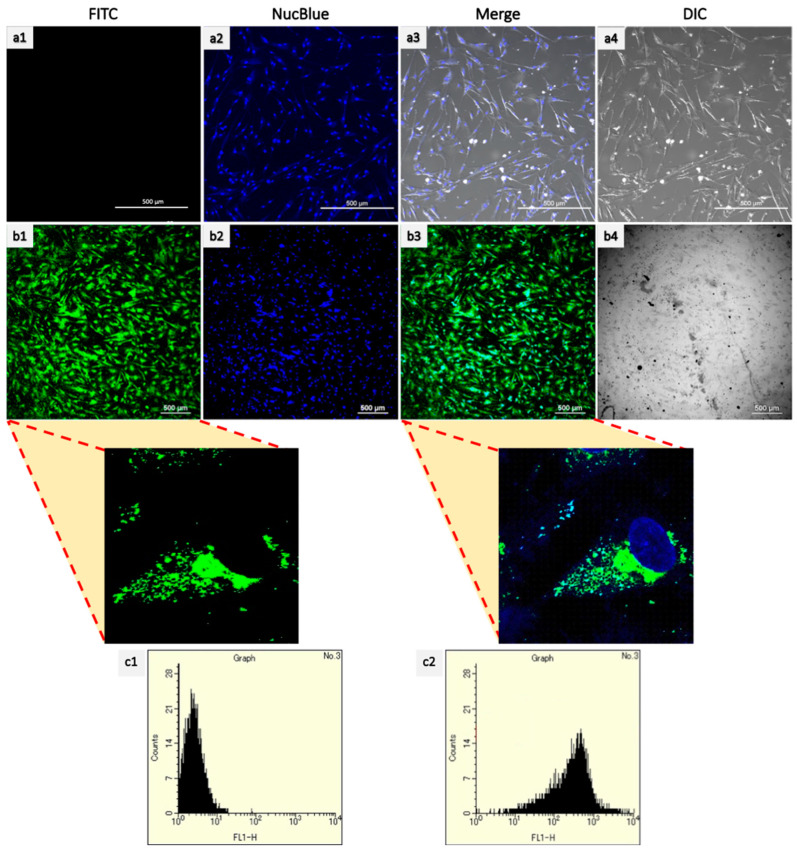
Cellular Uptake of FITC-loaded PLGA nanoparticles using confocal microscopy and flow cytometry. The nanoparticles were labeled with FITC (green). The cells were incubated with PLGA-FITC NPs (**b1**–**b4**) for 4 h to evaluate cellular internalization of the polymer into the cell compared with the control (**a1**–**a4**). FITC particle internalization using flow cytometry (**c1**,**c2**). (**c2**) shows particle uptake by HCF cells. (Scale bar = 20 and 500 μm).

**Figure 6 ijms-23-15179-f006:**
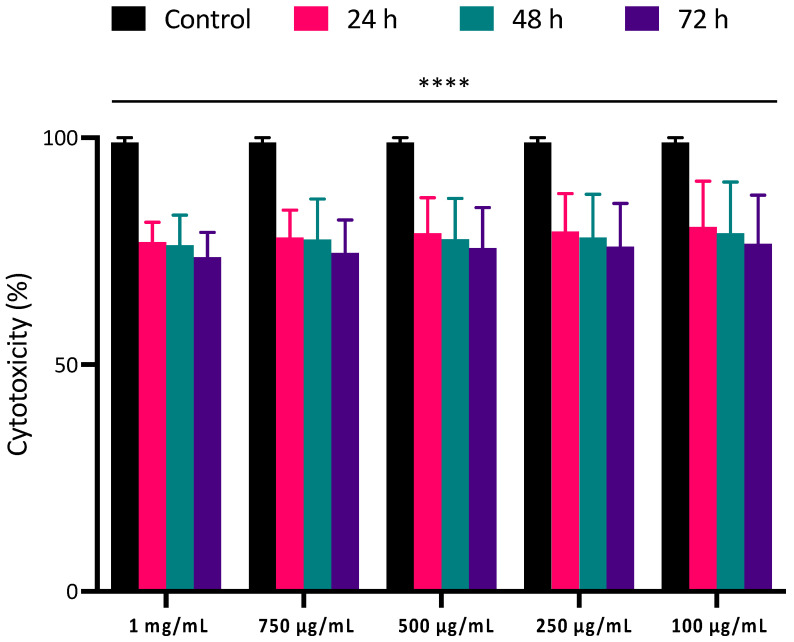
Cytocompatibility of PLGA void NPs was evaluated to assess the compatibility with HCF at a concentration of 1 mg/mL, 750 μg/mL, 500 μg/mL, 250 μg/mL 100 μg/mL. The ratios of PLGA NPs at all doses and time points demonstrated that the experiment was extremely statistically significant, with *p* < 0.05 (**** *p* ≤ 0.0001). Experiment was extremely statistically significant, with *p* < 0.05 **** *p* ≤ 0.0001).

**Figure 7 ijms-23-15179-f007:**
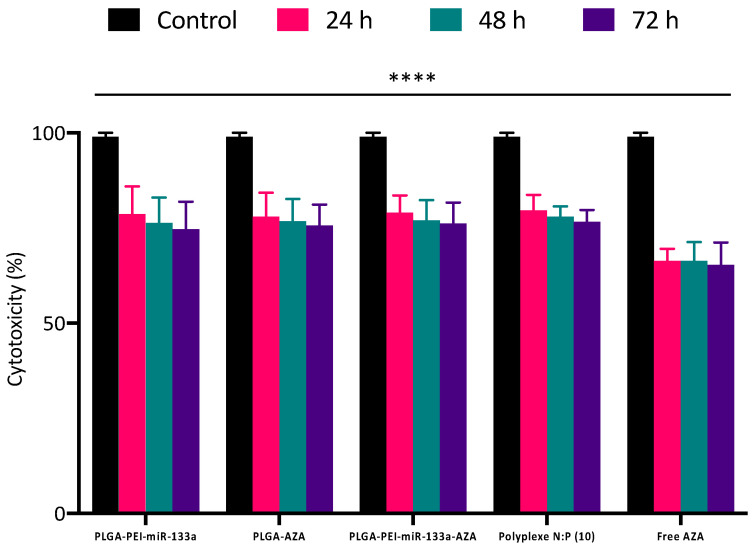
Cell viability was evaluated to assess the compatibility of HCF treated with PLGA NPs PLGA-PEI-miR-133a, PLGA-AZA, and PLGA-PEI-miR-133a-AZA at a concentration of 100 μg/mL in comparison with miR-133a polyplexes (50 nM was used to for complex with PEI at an N:P ratio 10), free drug (1 mg/mL) and control for 24 h, 48 h, and 72 h. The cell viability of all PLGA NPs was 80% viable at all time points. The ratios of PLGA NPs at all doses and time points demonstrated that the experiment was extremely statistically significant, with *p* < 0.05 (**** *p* ≤ 0.0001). Experiment was extremely statistically significant, with *p* < 0.05 (**** *p* ≤ 0.0001).

**Figure 8 ijms-23-15179-f008:**
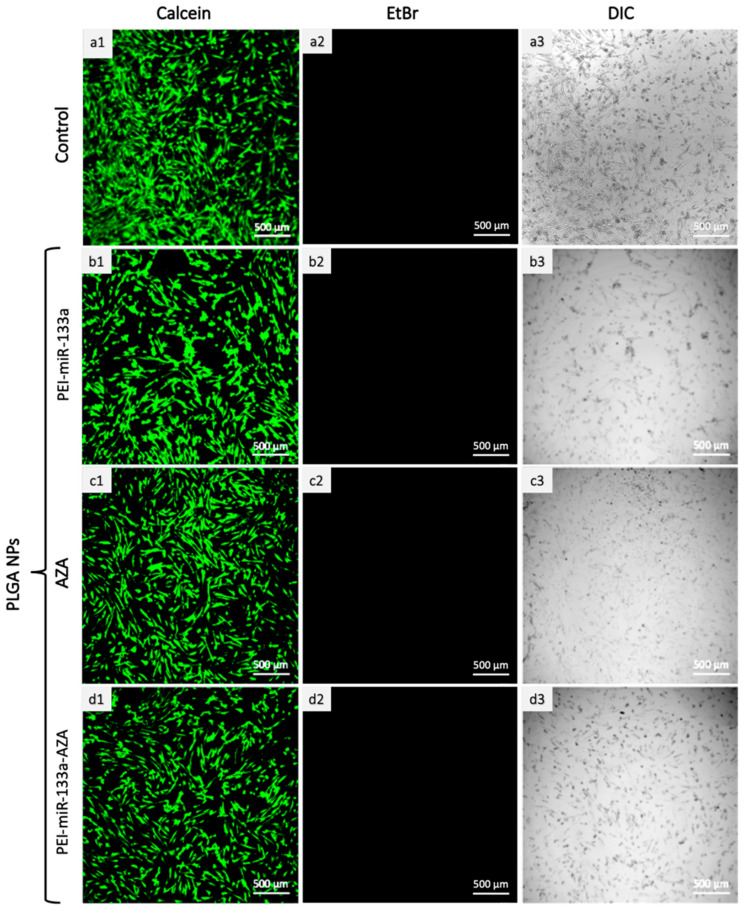
Live-dead analysis to visualize the cytocompatibility of HCF cells treated with PLGA-PEI-miR-133a (**b1**–**b3**), PLGA-AZA (**c1**–**c3**), and PLGA-PEI-miR-133a-AZA (**d1**–**d3**). The cells treated with PLGA NPs encapsulated with PEI-miR-133a (**b1**–**b3**), AZA (**c1**–**c3**), and PEI-miR-133a-AZA (**d1**–**d3**) show live cells similar to that of the control group (**a1**–**a3**). The green color (calcein) stained cells represent the live cells, while the red color (EtBr) stained cells represent dead cells that are not present. Thus, confirming that PLGA NPs exhibit minimal toxicity and are metabolically active. (scale bar = 500 µm).

**Figure 9 ijms-23-15179-f009:**
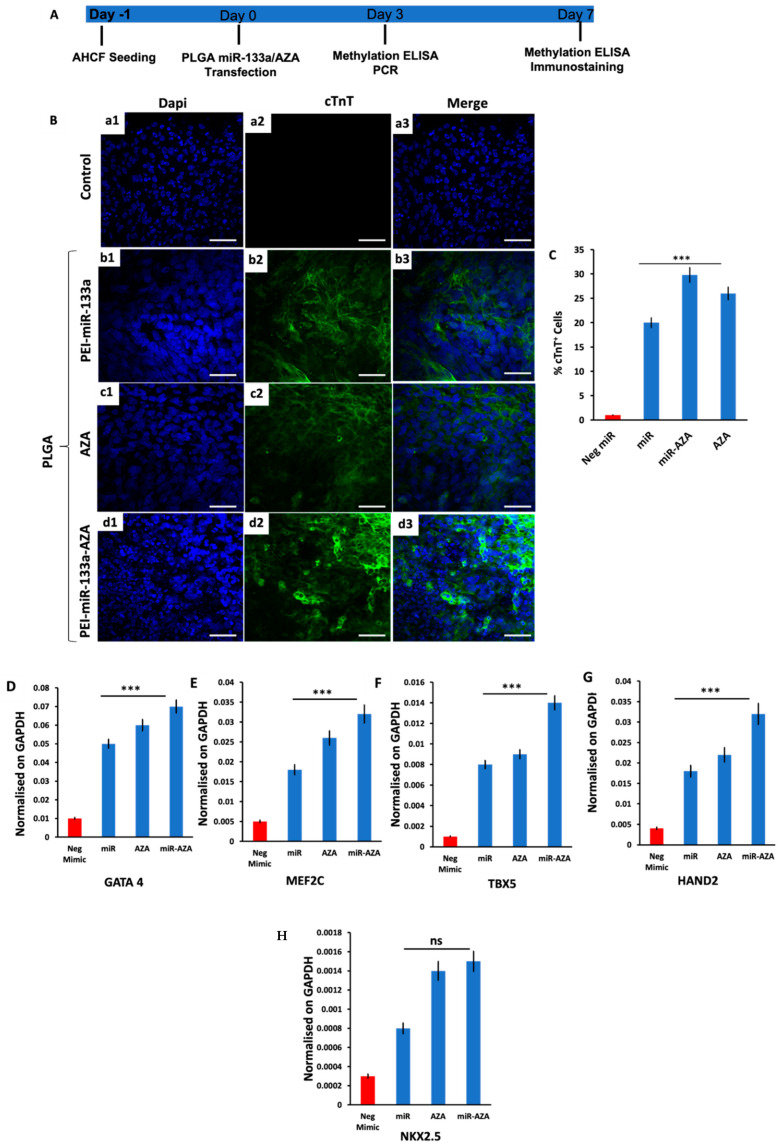
MiR-133a, AZA, and MiR-133a-AZA-transfected AHCFs show cTnT expression and increased cardiac transcription factor expression. (**A**) Representative scheme of experimental design. AHCFs were transfected with MiR-133a, AZA, and MiR-133a-AZA or negative control (negmiR). The acquisition of cardiomyocyte-associated features was evaluated after 3 days (Methylation ELISA and qpcr for cardiac transcription factors), 7 days (Immunostaining for cardiomyocyte marker and Methylation ELISA). (**B**) Immunostaining analysis to evaluate HCF cells treated with PLGA-PEI-miR-133a (b1–b3), PLGA-AZA (c1–c3), and PLGA-PEI-miR-133a-AZA (d1–d3). The cells treated with both miR-133a, and AZA show relatively more cTnT-positive cells when compared with untreated cells (control) (a1–a3), miR-133a, and AZA alone. (**C**) cTnT positive cell count expressed in percentage. (**D**–**H**) Gene expression of cardiac transcription factors. The expression of *GATA4*, *MEF2C*, *TBX5*, *HAND2*, and *NKX2.5* was evaluated by qPCR 7 days post-transfection in AHCFs transfected with MiR-133a, AZA, and MiR-133a-AZA (blue) or negmiR (red). Data are representative of three independent experiments, each performed in triplicate. Stated *p*-value is versus negmiR controls. (scale bar = 50 µm). The experiment was statistically significant, with *p* < 0.05 (*** *p* ≤ 0.001).

**Figure 10 ijms-23-15179-f010:**
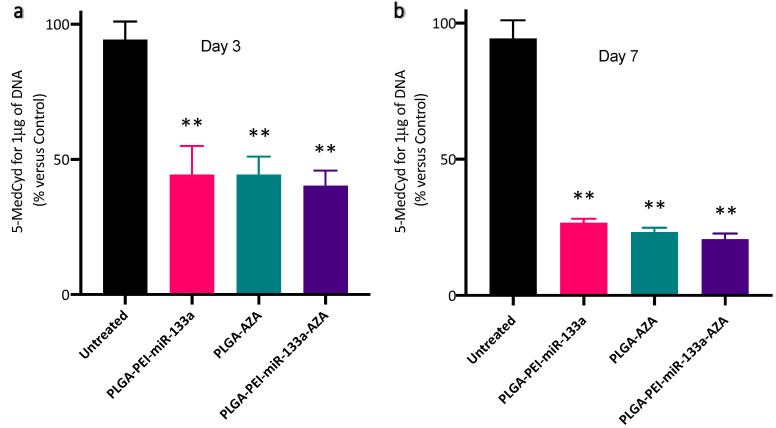
Effect of PLGA NPs (PEI-miR-133a, AZA, and PLGA-PEI-miR-133a-AZA) treatment on HCF cells. (**a**) Quantification of 5′-methyl-2′-deoxycytidine (5-MedCyd) in the DNA sample after treatment with different PLGA NP formulations in HCF cells using ELISA on day 3 (**b**) Quantification of 5′-methyl-2′-deoxycytidine (5-MedCyd) in the DNA sample after treatment with different PLGA NP formulation in HCF cells using ELISA on day 7. ** *p* ≤ 0.01.

**Table 1 ijms-23-15179-t001:** Size distribution, polydispersity index, zeta potential, yield (%), encapsulation efficiency (%) of PLGA NPs.

Sample	Z-Average Diameter (nm)	Polydispersity Index (PDI)	Zeta Potential (mV)	Yield (%)	Encapsulation Efficiency (%)
PLGA void	186 ± 3.1	0.063 ± 0.06	−15.6 ± 0.04	37.13%	-
PEI miRNA	124 ± 0.6	0.301 ± 0.04	27.35 ± 0.03		-
PLGA AZA	298 ± 2.0	0.034 ± 0.03	−15.8 ± 0.86	37.35%	96%
PLGA AZA-miRNA	298 ± 2.4	0.031 ± 0.02	−15.7 ± 0.65	38.15%	97%
PLGA-miRNA	298 ± 2.6	0.032 ± 0.02	−15.6 ± 0.34	37.35%	96%

## Data Availability

The data supporting the findings of this study are available from the corresponding author upon reasonable request.

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
