# Peer review of "Direct Cardiac Epigenetic Reprogramming through Codelivery of 5′Azacytidine and miR-133a Nanoformulation"

_ijms, 2022, doi:10.3390/ijms232315179_

Round 1

Reviewer 1 Report

The authors aimed to reprogram cardiac fibroblasts into cardiomyocytes using a combination between 5'Aza and miR-133a incapsulated into a PLGA-PEI carrier. The topic is interesting but some concern should be clarified

1) In figure 9 the authors reported the detecton of TnT by immunofluorescence. In their picture TnT has a predominant nuclear localization, but since it is a protein associate with the sarcomere, it should localize into the cytoplasm. I think that the figure 9 is not very convincing. I would also suggest to analyse the expression of other cardiac markers, such as NKX2.5, Sarcomeric actinin, SERCA, CACNA. Also performing a quantification of each marker expression.

2) Despite the authors reported that the treatment with 5AZA + PEI-miR133a enhanced the ihibition of DNA methylation  (line 55), I do not see any difference among the three experimental conditions.

3) in Figure 3, please add the % of agarose in the gel and the size of the ladder.

4) In paragraph 2.2.7 please explain which medium did you used in more details.

Author Response

Dear Editor,

We thank the respected Editor and the Editorial Board for their time and effort in considering our manuscript. We sincerely thank all the esteemed reviewers for their valuable comments and suggestions to help improve our manuscript. We appreciate the time they invested in evaluating our presentation and providing their expert opinions to enrich our work further. We have tried our sincere best to comply with the reviewers' suggestions and hope they will accept them. We have addressed the comments/suggestions of reviewers and have revised the manuscript appropriately. We genuinely hope our revised version manuscript will be appropriate for publication in "The MDPI International Journal of Molecular Sciences" after revision. The highlighted regions indicate changes made to the manuscript.

The Reviewer's comments and our response are as follows.

The authors aimed to reprogram cardiac fibroblasts into cardiomyocytes using a combination between 5'Aza and miR-133a encapsulated into a PLGA-PEI carrier. The topic is interesting but some concerns should be clarified

1) In figure 9 the authors reported the detecton of TnT by immunofluorescence. In their picture TnT has a predominant nuclear localization, but since it is a protein associate with the sarcomere, it should localize into the cytoplasm. I think that the figure 9 is not very convincing. I would also suggest to analyse the expression of other cardiac markers, such as NKX2.5, Sarcomeric actinin, SERCA, CACNA. Also performing a quantification of each marker expression.

We want to thank the reviewer for giving us this valuable opinion. Per your suggestion, we have performed the immunofluorescence again with the cTnT marker and quantified the marker expression. The changes have now been made in the revised submission.

2) Despite the authors reported that the treatment with 5AZA + PEI-miR133a enhanced the inhibition of DNA methylation(line 55), I do not see any difference among the three experimental conditions.

We want to thank the reviewer for pointing this out. We intended to say that, compared to day 3 (Figure 10 (b)), a significant reduction in 5-MedCyd was observed on day 7 (Figure 10 (b)). Further, on day 7, PLGA-5'AZA (23%) treated cells show a slightly reduced 5-MedCyd value compared to the PLGA-PEI-miR-133a (26%). While the 5-MedCyd value of PLGA-PEI-miRNA and PLGA-5'AZA were significantly reduced, the co-delivery of miR-133a and 5'AZA (20%) treated cells showed further hypomethylation, which could be due to the inhibition of DNA methylation activity by both miR-133a and 5'AZA. The same is mentioned in the revised submission.

 3) in Figure 3, please add the % of agarose in the gel and the size of the ladder.

We want to thank the reviewer for giving us this kind suggestion. We have used 1% agarose gel for the gel electrophoresis. The changes have now been made in the revised submission.

4) In paragraph 2.2.7 please explain which medium did you used in more details.

Human Cardiac Fibroblasts (HCF) were cultured at a cell density of 5x106 cells in a specialized cardiac fibroblast growth media and Human Cardiomyocytes (HCM) were cultured in cardiomyocyte media respectively at 37ºC in a 5% CO2 incubator. The cells were subcultured every 6-7 days or until confluent.

Reviewer 2 Report

In the study of Muniyandi et al., the authors investigated the potential of direct cardiac epigenetic reprogramming through codelivery 2 of 5’Azacytidine and miR-133a via nanoparticles. Comprehensive analysis and characterization was done on the biochemical components used in this study. However, the characterization of resulting human cardiomyocytes is not convincing. The directly reprogrammed human cardiomyocytes presented in Figure 9 show no visible sarcomere striation, which should be visible when using the cardiac sarcomere marker cTnT. Therefore, I have the following suggestions for the authors:

·       Please specify in the results or discussion part what kind of human cardiomyocytes were generated in this study (fetal-like, mature or adult cardiomyocytes)?

·       Furthermore, please add higher magnification or quality images in which sarcomere striation is visible.

·       Additionally, please perform a molecular analysis (qRT-PCR or Western Blot) of generated human cardiomyocytes for cardiac and fibroblast markers to provide a stronger claim that cardiomyocytes were successfully generated using the here presented differentiation approach.

·       Please shorten the introduction and lengthen the discussion. Many points are discussed in the result parts that can be moved to the discussion.

Major comments:

·       2.2.7 Materials: Please specify if the human cardiomyocytes (HCM) mentioned in this part are commercially derived or the directly reprogrammed cells. Furthermore, please add in the materials part the culture medium that was used for HCMs.

·       Please add a schematic presentation of the study protocol used for direct reprogramming (timepoint of cardiac induction, experimental timepoints, etc.)

·       Figure 5 and Figure S3: Please complete legends for all assigned images (missing: b1-b4; a2-3 and b2-3, respectively)

·       Figure 6, 7, 10 and S4: Please add statistical analysis

·       Figure 9: Please add quantification of cTnT+ cells for all investigated groups.

Minor comments:

·       To prevent any confusion on which cell type was used please emphasize in the introduction that commercial human cardiac fibroblasts were used for this study.

·       Line 61: “…leading to an impaired reprogramming process was.”

·       Line 250: “…instructions on the Live/Dead cytotoxicity kit.” -> Please add manufacturer and catalog number

·       Line 276: “…Genomic DNA was extracted with Purelink® Genomic DNA kit according… .” .” -> Please add manufacturer and catalog number

·       Line 294: “._The morphology of PEI-miR-133a…” -> add a space at the beginning of the sentence

o   Same for Line 297

o   Please check the manuscript for spacing errors in the text, there are several where there is too much or not no spacing in the text.

·       Line 491-500: Please shorten this part or move parts of the literature discussed into the discussion.

Author Response

Dear Editor,

We thank the respected Editor and the Editorial Board for their time and effort in considering our manuscript. We sincerely thank all the esteemed reviewers for their valuable comments and suggestions to help improve our manuscript. We appreciate the time they invested in evaluating our presentation and providing their expert opinions to enrich our work further. We have tried our sincere best to comply with the reviewers' suggestions and hope they will accept them. We have addressed the comments/suggestions of reviewers and have revised the manuscript appropriately. We genuinely hope our revised version manuscript will be appropriate for publication in "The MDPI International Journal of Molecular Sciences" after revision. The highlighted regions indicate changes made to the manuscript.

The Reviewer's comments and our response are as follows.

 Reviewer: 2

In the study of Muniyandi et al., the authors investigated the potential of direct cardiac epigenetic reprogramming through codelivery 2 of 5’Azacytidine and miR-133a via nanoparticles. Comprehensive analysis and characterization was done on the biochemical components used in this study. However, the characterization of resulting human cardiomyocytes is not convincing. The directly reprogrammed human cardiomyocytes presented in Figure 9 show no visible sarcomere striation, which should be visible when using the cardiac sarcomere marker cTnT. Therefore, I have the following suggestions for the authors:

1) Please specify in the results or discussion part what kind of human cardiomyocytes were generated in this study (fetal-like, mature or adult cardiomyocytes)?

Thank you for the reviewer’s comments. We have generated more mature or adult-like cardiomyocytes.

2) Furthermore, please add higher magnification or quality images in which sarcomere striation is visible.

Thanks for the reviewer’s comments, we have included the images showing sarcomere striation in the edited manuscript.

3) Additionally, please perform a molecular analysis (qRT-PCR or Western Blot) of generated human cardiomyocytes for cardiac and fibroblast markers to provide a stronger claim that cardiomyocytes were successfully generated using the here presented differentiation approach.

We would like to thank the reviewer for giving us this kind suggestion. We have performed qPCR to claim that we have successfully generated cardiomyocyte-like cells. The changes have now been made in the revised submission.

4) Please shorten the introduction and lengthen the discussion. Many points are discussed in the result parts that can be moved to the discussion.

We would like to thank the reviewer for giving us this kind suggestion. We have tried our best to edit the introduction in the revised submission.

Major comments:

5) 2.2.7 Materials: Please specify if the human cardiomyocytes (HCM) mentioned in this part are commercially derived or the directly reprogrammed cells. Furthermore, please add in the materials part the culture medium that was used for HCMs.

We would like to thank the reviewer for giving us this kind suggestion. The HCMs used in this study were commercially derived. The changes have now been made in the revised submission.

6) Please add a schematic presentation of the study protocol used for direct reprogramming (timepoint of cardiac induction, experimental timepoints, etc.)

We would like to thank the reviewer for giving us this kind suggestion. We have made a schematic illustration of the experimental design.  The changes have now been made in the revised submission.

7) Figure 5 and Figure S3: Please complete legends for all assigned images (missing: b1-b4; a2-3 and b2-3, respectively)

We would like to thank the reviewer for giving us this kind suggestion. The changes have now been made in the revised submission.

8) Figure 6, 7, 10 and S4: Please add statistical analysis

We would like to thank the reviewer for giving us this kind suggestion. The statistical analysis has been added to the revised submission.

9) Figure 9: Please add quantification of cTnT+ cells for all investigated groups.

We would like to thank the reviewer for giving us this kind suggestion. The cTnT positive quantification graph has now been added to the revised manuscript.

Minor comments:

10) To prevent any confusion on which cell type was used please emphasize in the introduction that commercial human cardiac fibroblasts were used for this study.

We would like to thank the reviewer for giving us this kind suggestion. The changes have now been made in the revised submission.

11) Line 61: “…leading to an impaired reprogramming process was.”

We would like to thank the reviewer for giving us this kind suggestion. The changes have now been made in the revised submission.

11) Line 250: “…instructions on the Live/Dead cytotoxicity kit.” -> Please add manufacturer and catalog number.

We would like to thank the reviewer for giving us this kind suggestion. The changes have now been made in the revised submission.

12) Line 276: “…Genomic DNA was extracted with Purelink® Genomic DNA kit according… .” .” -> Please add manufacturer and catalog number

We would like to thank the reviewer for giving us this kind suggestion. The changes have now been made in the revised submission.

13) Line 294: “._The morphology of PEI-miR-133a…” -> add a space at the beginning of the sentence

o   Same for Line 297

We would like to thank the reviewer for giving us these kind suggestions. The changes have now been made in the revised submission.

14) Please check the manuscript for spacing errors in the text, there are several where there is too much or not no spacing in the text.

We would like to thank the reviewer for giving us this kind suggestion. The changes have now been made in the revised submission.

15) Line 491-500: Please shorten this part or move parts of the literature discussed into the discussion.

We would like to thank the reviewer for giving us this kind suggestion. The changes have now been made in the revised submission.

Round 2

Reviewer 1 Report

The authors addessed all the reviewer's comments

Reviewer 2 Report

I think my comments have been adequately answered and therefore I have nothing more to add.